# Association between Different Animal Protein Sources and Liver Status in Obese Subjects with Non-Alcoholic Fatty Liver Disease: Fatty Liver in Obesity (FLiO) Study

**DOI:** 10.3390/nu11102359

**Published:** 2019-10-03

**Authors:** Gregorio Recaredo, Bertha Araceli Marin-Alejandre, Irene Cantero, J. Ignacio Monreal, José Ignacio Herrero, Alberto Benito-Boillos, Mariana Elorz, Josep A. Tur, J. Alfredo Martínez, M. Angeles Zulet, Itziar Abete

**Affiliations:** 1Department of Nutrition, Food Sciences and Physiology and Centre for Nutrition Research, Faculty of Pharmacy and Nutrition, University of Navarra, 31008 Pamplona, Spain; grecaredo@alumni.unav.es (G.R.); bmarin.1@alumni.unav.es (B.A.M.-A.); icgonzalez@unav.es (I.C.); jalfmtz@unav.es (J.A.M.); 2Navarra Institute for Health Research (IdiSNA), 31008 Pamplona, Spain; jimonreal@unav.es (J.I.M.); iherrero@unav.es (J.I.H.); albenitob@unav.es (A.B.-B.); marelorz@unav.es (M.E.); 3Clinical Chemistry Department, University Clinic of Navarra, University of Navarra, 31008 Pamplona, Spain; 4Liver Unit, Clinica Universidad de Navarra, 31008 Pamplona, Spain; 5Centro de Investigación Biomédica en Red de Enfermedades Hepáticas y Digestivas (CIBERehd), 28029 Madrid, Spain; 6Department of Radiology, Clinica Universidad de Navarra, 31008 Pamplona, Spain; 7Biomedical Research Centre Network in Physiopathology of Obesity and Nutrition (CIBERobn), Instituto de Salud Carlos III, 28029 Madrid, Spain; pep.tur@uib.es; 8Research Group on Community Nutrition and Oxidative Stress, University of Balearic Islands, 07122 Palma, Spain

**Keywords:** obesity, NAFLD, fatty liver, red meat, processed meat, fish, ferritin, iron

## Abstract

Non-alcoholic fatty liver disease (NAFLD) is considered the hepatic manifestation of metabolic syndrome. Obesity and unhealthy dietary habits are described as risk factors for NAFLD. The aim of this study was to investigate the association between the consumption of different animal protein sources and hepatic status in NAFLD adults. A total of 112 overweight/obese participants with NAFLD from Fatty Liver in Obesity (FLiO) study were evaluated at baseline. Diet, body composition, and biochemical variables were evaluated. Hepatic status was also assessed by Magnetic Resonance Imaging, ultrasonography, and elastography. Red meat consumption showed a positive relationship with liver iron content (r = 0.224; *p* = 0.021) and ferritin concentration (r = 0.196; *p* = 0.037). Processed meat consumption exhibited a positive association with liver iron content (r = 0.308; *p* = 0.001), which was also found in the quantile regression (β = 0.079; *p* = 0.028). Fish consumption was related with lower concentration of ferritin (r = −0.200; *p* = 0.034). This association was further evidenced in the regression model (β = −0.720; *p* = 0.033). These findings suggest that the consumption of different animal protein sources differentially impact on liver status in obese subjects with NAFLD, showing fish consumption as a healthier alternative for towards NAFLD features.

## 1. Introduction

Obesity is a chronic disease, characterized by a low-grade inflammatory systemic manifestation due to the interaction of multiple factors such as environmental and genetic background among others [1,2]. Obesity rates have increased gradually becoming an important clinical and public health problem. It is expected that 1.12 billion adults will be obese in the next 10 years [3]. Metabolic comorbidities including type 2 diabetes, insulin resistance, dyslipidemia, dysbiosis, metabolic syndrome, and non-alcoholic fatty liver disease (NAFLD) have risen along with the growing incidence of obesity [2,3]. Non-alcoholic fatty liver disease is considered part of the metabolic syndrome [4,5]. Its prevalence ranges from 6% to 35% in the general population [4]. NAFLD is characterized by an excessive lipid accumulation in the liver of people who drink little or no alcohol. The most common form of NAFLD is a non-serious condition called fatty liver. This condition may progress to a more serious state named non-alcoholic steatohepatitis (NASH), where fat accumulation is associated with liver cell inflammation. NASH is a potentially severe condition that might lead to liver cirrhosis and patients with cirrhosis may eventually require a liver transplant [5,6,7,8]. Obesity is considered the primary cause of fatty infiltration of the liver; however, less is known about NASH development. Current investigations tried to identify factors that may contribute to the development and progression of the disease. Unhealthy Western lifestyle might play a role in the development and progression of NAFLD, due to the high consumption of red meat, processed and preserved meat, snacks, refined grains, pastries, sugar-sweetened beverages, saturated fat, and lack of physical activity [5,9,10,11]. Global meat consumption has increased over the past decades and harmful effects have been reported particularly of red and processed meat [9]. Observational studies have shown that red meat consumption seems to be associated with insulin resistance, type 2 diabetes, metabolic syndrome, and oxidative stress [9,12,13]. In the same way, processed meat has been related with metabolic syndrome, type 2 diabetes, and high risk of mortality from all causes [14]. However, data regarding the association between different animal protein sources and non-alcoholic fatty liver disease are currently lacking or controversial [9,15,16]. In this context, the aim of this study was to evaluate the relationship between different animal protein sources and hepatic status in obese patients with NAFLD.

## 2. Materials and Methods

### 2.1. Study Participants

The current study included 112 (65 Male and 47 Female) overweight/obese (BMI ≥ 25 to <40 kg/m^2^) adults between 40 and 80 years old and with ultrasound-confirmed liver steatosis. The study population corresponds to patients with NAFLD from the Fatty Liver in Obesity (FLiO) study evaluated at baseline. The exclusion criteria were defined as presence of known liver disease other than NAFLD, ≥3 kg body weight loss prior to visit, significant alcohol consumption (>21 standard drinks per week in men and >14 standard drinks per week in women), endocrine disorders (hyperthyroidism or uncontrolled hypothyroidism), pharmacological treatments (immunosuppressants, cytotoxic agents, systemic corticosteroids or the use of steatogenic drugs or medication that alter liver function), existence of active autoimmune diseases or requiring pharmacological treatment, use of weight-loss medication, and presence of severe psychiatric disorders. This information was declared by the subjects in the clinical interview before their enrollment in the study. All study procedures were in accordance with the Declaration of Helsinki and were approved by the Research Ethics Committee of the University of Navarra (ref. 54/2015). The study protocol was properly registered in www.clinicaltrails.gov (FLiO: Fatty Liver in Obesity study; NCT03183193). All participants gave written informed consent.

### 2.2. Anthropometric, Body Composition, and Biochemical Measurements

Anthropometric measurements were assessed in fasting conditions as described elsewhere [17]. Body weight, waist circumference (WC), body composition (fat mass and lean mass) by dual-energy X-ray absorptiometry (Lunar iDXA, encore 14.5, Madison, WI, USA), and blood pressure (Intelli Sense. M6, OMRON Healthcare, Hoofddorp, The Netherlands) were determined at the Metabolic Unit of the University of Navarra. Body Mass Index (BMI) was calculated as the body weight divided by the squared height (kg/m^2^). Blood samples were properly collected (after 8–10 h of fasting), centrifugated (15 min; 3500 rpm; 5 °C), aliquoted (0.5 mL Eppendorf), and stored at −80 °C for further analyses. Serum glucose, lipid profile parameters: Total cholesterol (TC); high-density lipoprotein cholesterol (HDL-c); low density lipoprotein cholesterol (LDL-c), and triglycerides (TG) as well as liver enzymes: aspartate aminotransferase (AST), alanine aminotransferase (ALT), gamma glutamyl transferase (GGT) concentrations were determined on an autoanalyzer (Pentra C-200; HORIBA ABX, Madrid, Spain) following the manufacturer’s instructions. Insulin concentrations were determined using specific Enzyme-Linked ImmunoSorbent Assay (ELISA) kits (Demeditec; Kiel-Wellsee, Germany) in another autoanalyzer (Triturus; Grifols, Barcelona, Spain). The homeostatic model assessment of insulin resistance: HOMA-IR = (insulin (µU/mL) × glucose (mmol/L))/22.5, was used to estimate insulin resistance. Ferritin serum levels were analyzed by an external certified laboratory (Eurofins Megalab S.A, Madrid, Spain) using a Chemiluminescent Microparticle Immunoassay (CMIA) technology as described elsewhere (Abbott Architect Ferritin Assay).

### 2.3. Lifestyle Assessment: Dietary and Physical Activity

Dietary intake of participants was collected with a semi-quantitative 137-item food frequency questionnaire (FFQ), validated in Spain [18]. For each item, a typical portion size was included, and consumption frequencies were registered in nine categories that ranged from “never or almost never” to “≥6 times/day”. Energy and Nutrient intake was calculated as frequency multiplied by nutrient composition of specified portion size for each food item, using a computer program based on available information in Spanish food composition tables [19]. For the purpose of the study, red meat was defined as meat from mammals without preservation methods: beef, pork, lamb, liver, and viscera [20]. Meat that has been transformed through salting, smoking, or preservation techniques, was defined as processed meat: bacon, burger, ham, pâté, cold meats, meatballs, and sausages [9]. White meat included chicken, turkey, rabbit, and hare. Fish was defined as white fish, fatty fish, salted fish, canned fish, clam, squid and shellfish, and canned shellfish. Physical activity was evaluated using the validated Spanish version of the Minnesota Leisure-Time Physical Activity Questionnaire [21].

### 2.4. Liver Status Assessment

Liver status was assessed under fasting conditions by qualified staff at the Clinic University of Navarra. The presence of hepatic steatosis was determined by means of Ultrasonography (Siemens ACUSON S2000 and S3000) in accordance with previously described methodology [6]. In addition, liver stiffness was evaluated by Acoustic Radiation Force Impulse (ARFI) elastography (Siemens ACUSON S2000 and S3000) and transient elastography through FibroScan^®^ (Echosens, Paris, France). Ten valid ARFI measurements were performed in each patient [22] to estimate the median value of liver stiffness. The same qualified radiologist from the department of Ultrasonography and Radiology executed all the ultrasonographic evaluations. In order to quantify the fat and iron content of the liver as well as the hepatic volume, a Magnetic Resonance Imaging (MRI) (Siemens Aera 1.5 T) was also performed to each participant following standardized procedures [6]. The MRI assessment of liver fat and iron content was performed through the Dixon method.

### 2.5. Statistical Analyses

Variable distribution was assessed by means of the Shapiro–Wilk test. Data were presented as mean ± standard deviation. Animal protein variables were categorized according to medians of consumption (g/day). The cut off points of each animal protein source were the following: red meat <53.7 g/day and ≥53.7 g/day; processed meat <42.5 g/day and ≥42.5 g/day; white meat <73.05 g/day and ≥73.05 g/day; fish <88.5 g/day and ≥88.5 g/day. Differences between groups (< and ≥ of the median) were assessed by the Student’s *t*-test and the Mann–Whitney U test according to the data distribution. The relationship between liver status variables and animal protein consumption was assessed by the Pearson’s correlation coefficient or the Spearman’s rho (*p*). Non-parametric variables were analyzed by quantile regression models and were adjusted for potential confounders such as age, sex, body mass index, energy intake, and physical activity. Confidence intervals were used to describe the regression coefficient (β) values. Analyses were performed using 12.0 (Stata Corp, College Station, TX, USA). All *p*-values were two-tailed. Values of *p* < 0.05 were considered statistically significant.

## 3. Results

The average age of participants was 51 ± 9 years. Men and women were equally distributed among medians of protein consumption. No significant differences were observed in animal protein intake between sexes. Descriptive characteristics of the population according to the median of animal protein consumption are reported in Table 1. Participants with higher red meat consumption (≥53.7 g/day) had significantly greater concentrations of triglycerides, alanine aminotransferase, and ferritin (*p* < 0.05 for all comparisons) than those with lower values of red meat consumption. Subjects with higher red meat intake showed significantly lower concentrations of HDL-c (*p* = 0.014) compared to those with lower consumption. Liver iron content was increased in participants above the median of red meat consumption compared to those below the median, but no significant differences were observed (*p* = 0.067). Variables related to body composition did not show statistical differences between subjects below and above the median of red meat consumption.

Significant differences were observed in lipid profile between consumers above and below the median of processed meat consumption. LDL-c concentration was significantly higher (*p* = 0.044) while HDL-c was lower (*p* = 0.044) in participants with higher processed meat intake (Table 1). Subjects above the median of processed meat consumption showed greater liver iron content (*p* = 0.012) and lower liver stiffness values (*p* = 0.029). No relevant differences were observed in body composition, cardiometabolic risk factors, and liver status variables between participants below and above the median of white meat and fish consumption.

To assess the relationship between different animal protein consumption and hepatic status, a correlation analysis was applied. No statistical associations were observed between animal protein consumption and liver fat content: red meat (r = 0.119; *p* = 0.223), processed meat (r = 0.023; *p* = 0.815), white meat (r = −0.040; *p* = 0.681), and fish intake (r = −0.022; *p* = 0.822). Red (r = 0.024; *p* = 0.021) and processed meat (r = 0.308; *p* = 0.001) consumption were positively associated with liver iron content (Figure 1), while no significant associations were observed with white meat (r = −0.029; *p* = 0.767) and fish consumption (r = −0.054; *p* = 0.585). On the other hand, red meat (r = 0.196; *p* = 0.037) consumption was associated with ferritin levels, while fish (r = −0.200; *p* = 0.034) intake showed a negative relationship (Figure 2). Processed meat (r = 0.108; *p* = 0.254) and white meat (r = 0.015; *p* = 0.869) consumption were not significantly associated to ferritin levels (Figure 2).

A quantile regression analysis was carried out to assess the influence of different animal protein consumption on hepatic status. Fish consumption showed a negative association with ferritin concentration (Table 2) (β = −0.720; *p* = 0.033; 95% CI = −1.383; −0.058). Processed meat consumption was positively associated with liver iron content (Table 3) (β = 0.079; *p* = 0.028; 95% CI = 0.028; 0.150), after adjusting by sex, age, body mass index, energy intake (kcal), and physical activity (METs-min/week).

## 4. Discussion

The present study evaluated the possible association between the consumption of different animal protein sources with body composition, cardiometabolic risk factors, and hepatic status in overweight or obese subjects with ultrasonography proven liver steatosis. Red and processed meat were positively associated with liver iron content and regression analysis showed that processed meat consumption was involved in the variability of hepatic iron content even after adjustment by potential confounders. Additionally, red meat showed a positive association with ferritin concentration, while fish consumption was inversely associated with ferritin levels.

Unhealthy dietary habits are described as risk factors in the development of NAFLD [11]. Poor adherence to healthy dietary patterns and the shift to Western patterns might play an important role in the development of NAFLD [10,11]. The consumption of foods that characterize Western dietary patterns such as soft drinks, fructose, red and processed meats, and fast foods have shown to have detrimental health consequences promoting the development of obesity and associated comorbidities such as NAFLD [9,23,24]. Concretely meat consumption has steadily increased worldwide, and red and processed meat represent the majority of meat intake [25]. Meat contains saturated fatty acids (SFA) and cholesterol which have been described as important risk factors for cardiometabolic diseases, including NAFLD. More potentially harmful compounds such as heme-iron, sodium, and preservatives [9] are in the spotlight of research due to their possible negative effects on health. It is well known that high meat intake, specifically red and processed meat, are associated with higher prevalence of metabolic syndrome, impaired insulin sensitivity, type 2 diabetes, and oxidative stress [5,9,26]. Iron is present in foods as heme or non-heme iron. Heme-iron is found mainly in animal based foods, and the rest of the iron (animal or plant food) is non-heme iron [27]. Research studies have shown that meat or heme-iron intake is related to serum ferritin concentration, and ferritin levels have been associated with glucose and lipid profile alterations [15,27]. 

Excess circulating iron is potentially toxic, leading to the formation of reactive oxygen species, which can contribute to oxidative stress in NASH [28,29]. Increased serum ferritin level is closely related with insulin resistance and impaired glucose tolerance and NAFLD pathogenesis [30]. According to Willman et al., 2019, after a 6-month dietary intervention, a significant decrease on ferritin levels was found in participants corresponding to the “no red meat group” [31]. The reduction of ferritin levels was positively correlated with liver fat content improvement. Our results corroborate that meat consumption, specifically red meat, was associated with ferritin concentration which could be influencing the development of NAFLD. Other research studies observed a relationship between ferritin and elevated ALT, a surrogate of liver damage [30,32,33]. In this context, we observed that participants above the median of red meat consumption presented higher ferritin and ALT concentrations than those with lower red meat consumption.

Interestingly, a negative association between fish intake and ferritin concentration was observed in the present study. The underlying mechanism responsible for this relation cannot be clarified, however, it could be hypothesized that fish composition, mainly fatty fish, characterized by their Omega-3 PUFAs content, such as eicosapentaenoic acid (EPA) and docosahexaenoic acid (DHA) might exert beneficial effects on ferritin levels. Several studies have shown that omega-3 PUFAs are inversely associated with NAFLD, by decreasing proinflammatory molecules, serum triglycerides, and improving liver histology [10,34].

Ferritin is a protein implicated in iron metabolism and reflects the iron storage of the organism. However, in inflammatory conditions ferritin concentrations are increased and it is synthesized as an acute phase protein, acting as a pro-inflammatory molecule capable of inducing liver damage. [33,35,36,37]. The potential anti-inflammatory effect and antioxidant capacity of omega-3 PUFAs [38], might ameliorate ferritin pro-oxidant and pro-inflammatory activity, although the exact molecular pathways are yet to be elucidated.

The liver is the principal iron-storage organ and plays a central role in iron metabolism [4,35]. Increased hepatic iron concentrations are associated with metabolic alterations including insulin resistance and type 2 diabetes [15]. This condition is commonly known as dysmetabolic iron overload syndrome (DIOS) [15,39,40]. Mild-to-moderate hepatic iron accumulation is identified in NAFLD and other liver diseases. It has been suggested that one third of patients with NAFLD have DIOS [15,37]. Recent studies have shown that higher liver iron content is associated with the progression of NAFLD, including development of steatohepatitis, advanced fibrosis, and cirrhosis [15,41].

Magnetic resonance imaging is an emerging and non-invasive method for assessing liver iron and hepatic steatosis [6,15,39]. Our findings suggest that red and processed meat are partly associated with liver iron content. Quantile regression analyses corroborated the positive association between processed meat consumption and liver iron accumulation and the negative association between fish intake and ferritin concentration. McKay et al., 2018 reported a positive association between beef consumption and liver iron content assessed by MRI in a study population over than 9000 subjects [15]. Interestingly, other authors have described that iron absorption is increased in some patients with NAFLD, due to the overexpression of the divalent metal transporter 1 (DMT1) [29]. Due to the high bioavailability, heme-iron absorption rates are four to five times higher than non-heme iron, considered as the main contributor to stored body iron [42]. The detrimental effects of liver iron and ferritin on health have been described by several authors [29,30,42].

Contrary to our expectations, subjects with higher processed meat consumption registered lower liver stiffness values, while previous studies indicated that processed meat consumption was associated with an increased risk of chronic liver disease [43]. This result might be explained since transient elastography is a technique to assess the presence of advanced fibrosis or cirrhosis [44] and according to several authors, the cut-off point in order to determine liver fibrosis ranges from 6.5 to 8 kPa [6,45,46,47]. Our study population presented lower liver stiffness values, suggesting that participants did not present hepatic fibrosis. In this sense, it can be hypothesized that the infiltration of lipid droplets in the hepatocytes might promote structural changes in the liver, altering hepatic elasticity and disrupting the wave propagation through the liver registering contradictory results. Likewise, the interpretation of elastography results must be evaluated with caution to establish any possible association between liver stiffness and processed meat consumption. Potential limitations have been described for the use of transient elastography in NAFLD, related to the high failure due to invalid measurements in patients with high BMI and/or central obesity [28,46].

It is important to mention some limitations of the present study: firstly, other components of animal protein sources (such as vitamins) and cooking methods might also influence the observed results. Secondly, dietary data were collected by using self-reported information of the participants, and thus, our results are susceptible to some degree of bias. Thirdly, causal inference cannot be made due to the cross-sectional design of this investigation. On the other hand, some strengths can be mentioned as well, such as participants were carefully selected following exclusion and inclusion criteria to avoid a heterogeneous sample. Participants were well characterized and the methodology used to assess liver status was ultrasonography, ARFI, and MRI (a robust and accurate method to determine iron and fat concentration in liver) and to our knowledge, few studies have evaluated the association of different animal protein source consumption in subjects with NAFLD.

## 5. Conclusions

Processed meat consumption might influence hepatic iron concentration, while fish intake improved ferritin levels, both proposed as potential risk factors in the development and progression of NAFLD. These findings suggest that the consumption of different animal protein sources differentially impact on liver status of obese subjects with NAFLD.

## Figures and Tables

**Figure 1 nutrients-11-02359-f001:**
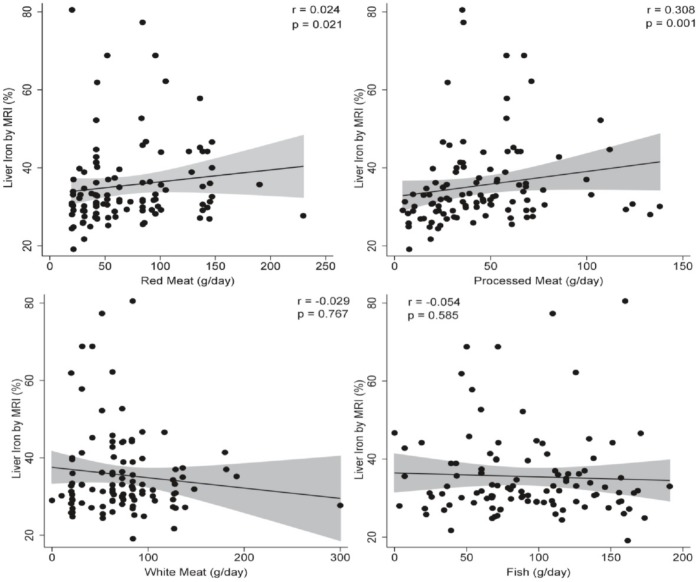
Correlation analyses to assess the relationship between different animal protein sources and liver iron content by MRI (%). *p* < 0.05 was considered statistically significant. MRI: magnetic resonance imaging.

**Figure 2 nutrients-11-02359-f002:**
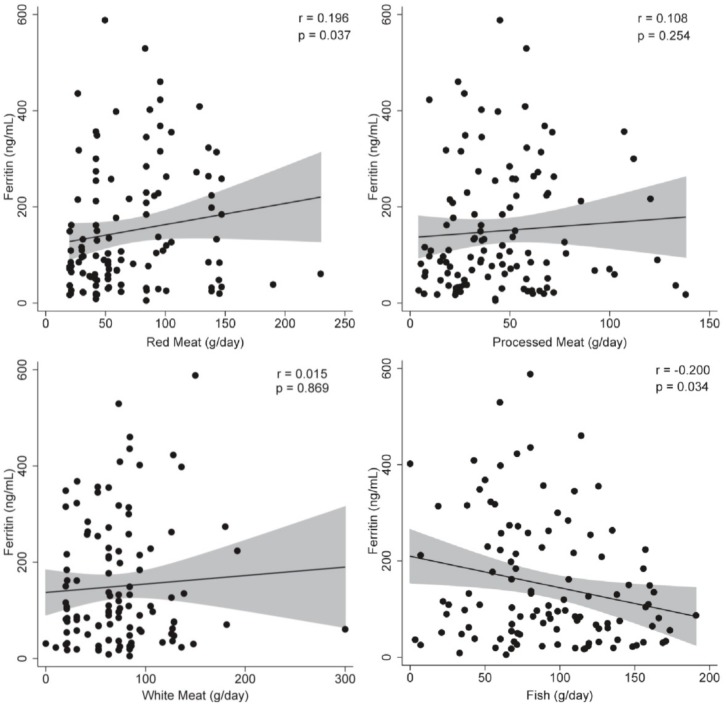
Correlations analyses to assess the relationship between different animal protein sources and ferritin levels. *p* < 0.05 was considered statistically significant.

**Table 1 nutrients-11-02359-t001:** Descriptive characteristics of study participants according to the median of animal protein consumption.

*n* = 112	Red Meat (g/day)		Processed Meat (g/day)		White Meat (g/day)		Fish (g/day)	
	<53.7	≥53.7	<42.5	≥42.5	<73.05	≥73.05	<88.5	≥88.5
**Body composition**								
BMI (kg/m^2^)	33.9 (4)	33.5 (3)	33.5 (3)	33.9 (4)	34.1 (4)	33.1 (3)	33.2 (4)	34.1 (4)
WC (cm)	110.0 (10)	109.5 (9)	109.8 (10)	109.8 (9)	111.2 (10)	107.9 (9)	108.1 (9)	111.4 (11)
Fat Mass (kg)	40.6 (9)	38.3 (8)	38.6 (7)	40.2 (10)	40.4 (8)	38.2 (9)	38.2 (8)	40.8 (9)
Lean Mass (kg)	51.9 (8)	54.2 (10)	53.0 (10)	53.0 (8)	54.4 (10)	51.3 (9)	52.9 (9)	53.1 (10)
**CMRF**								
Glucose (mg/dL)	110.0 (29)	107.9 (34)	108.0 (26)	109.9 (37)	112.0 (38)	104.9 (14)	109.3 (36)	108.6 (26)
TG (mg/dL)	129.3 (88)	146.3 (70) *	128.7 (75)	146.6 (83)	138.5 (78)	136.8 (82)	141.3 (74)	134.2 (85)
SPB (mmHg)	129.0 (16)	133.5 (12)	131.2 (16)	131.4 (12)	133.5 (15)	128.4 (12)	129.5 (13)	133.1 (15)
DBP (mmHg)	86.3 (11)	87.5 (6)	87.2 (10)	86.1 (7)	87.2 (9)	86.5 (8)	86.3 (8)	87.5 (9)
HDL-c (mg/dL)	54.6 (13)	49.5 (14) *	54.4 (14)	49.8 (14) *	52.3 (14)	51.7 (13)	51.5 (14)	52.6 (14)
LDL-c (mg/dL)	113.9 (31)	121.3 (38)	110.8 (38)	124.1 (31) *	120.2 (32)	114.3 (38)	116.2 (35)	119.1 (35)
TC (mg/dL)	193.6 (36)	200.2 (42)	190.3 (42)	203.3 (36)	199.6 (36)	193.3 (43)	196.1 (38)	197.8 (41)
HOMA-IR	5.5 (4.6)	4.9 (5.0)	5.1 (3.4)	5.3 (5.9)	5.5 (5.3)	4.8 (4.0)	5.2 (5.2)	5.2 (4.4)
HbA1c (%)	5.9 (1.0)	5.8 (1.0)	5.9 (0.9)	5.9 (1.2)	6.1 (1.3)	5.6 (0.3)	5.8 (1.1)	5.9 (1.0)
Insulin (U/mL)	19.2 (13)	17.2 (8)	18.3 (10)	18.2 (12)	18.4 (10)	18.0 (12)	18.1 (10)	18.4 (12)
**Liver Status**								
ALT (IU/L)	29.6 (15)	36.7 (20) *	34.6 (21)	31.7 (14)	34.2 (18)	31.8 (18)	35.7 (19)	30.6 (16)
AST (IU/L)	23.8 (11)	25.1 (8)	25.8 (12)	23.2 (7)	24.8 (10)	23.9 (9)	26.2 (12)	22.7 (7)
GGT (IU/L)	34.9 (24)	40.1 (27)	36.3 (24)	38.7 (27)	36.7 (26)	38.6 (24)	40.4 (27)	34.7 (24)
Ferritin (ng/mL)	119.2 (116)	181.0 (137) *	135.4 (119)	164.3 (139)	144.3 (116)	157.8 (146)	178.7 (147)	121.5 (104)
Liver Fat (%)	9.6 (11.1)	8.7 (7.8)	9.7 (9.6)	8.6 (9.7)	9.6 (10.4)	8.7 (8.6)	8.9 (9.8)	9.4 (9.5)
Liver Iron (%)	34.1 (11)	36.9 (11)	34.1 (12)	37.1 (10) *	36.8 (12)	33.9 (9)	35.6 (11)	35.4 (12)
MRI H.V (mL)	1799 (552)	1855 (444)	1749(461)	1906 (528)	1887 (575)	1752 (376)	1784 (436)	1870 (557)
LSF (kPa)	4.6 (1.8)	5.4 (2.2)	5.5 (2.2)	4.5 (1.8) *	5.2 (2.1)	4.8 (1.9)	4.7 (1.8)	5.3 (2.2)
ARFI (m/s)	1.9 (0.8)	1.7 (0.7)	1.8 (0.7)	1.9 (0.8)	1.8 (0.8)	1.9 (0.7)	1.8 (0.8)	1.8 (0.7)
**Steatosis Degree**								
Grade 1 (*n*) (%)	30 (53.5%)	32 (57.1%)	27 (49.1%)	35 (61.4%)	34 (53.1%)	28 (58.3%)	30 (53.5%)	32 (57.1%)
Grade 2 (*n*) (%)	19 (33.9%)	19 (33.9%)	23 (41.8%)	15 (26.3%)	21 (32.8%)	17 (35.4%)	20 (35.7%)	18 (32.1%)
Grade 3 (*n*) (%)	7 (12.5%)	5 (8.9%)	5 (9.1%)	7 (12.2%)	9 (14.1%)	3 (6.2%)	6 (10.7%)	6 (10.7%)

(Mean ± SD); * *p* < 0.05; BMI: body mass index; WC: waist circumference; CMRF: cardiometabolic risk factors; TG: triglycerides; SBP: systolic blood pressure; DPB: diastolic blood pressure; HDL-c: high-density lipoprotein cholesterol; LDL-c: low-density lipoprotein cholesterol; TC: Total cholesterol; HOMA-IR: homeostatic model assessment of insulin resistance; HbA1c: glycosylated hemoglobin; ALT: alanine aminotransferase; AST: aspartate aminotransferase; GGT: gamma-glutamyl transferase; MRI H.V: magnetic resonance imaging hepatic volume; LSF: liver stiffness; ARFI: acoustic radiation force impulse elastography.

**Table 2 nutrients-11-02359-t002:** Quantile regression models with Ferritin as the dependent variable and dietetic factors as independent variables.

Model 1					Model 2			
	β	(95% CI)		*p*-Value	β	(95% CI)		*p*-Value
Red Meat	0.349	(−0.271	0.970)	0.267	0.281	(−0.507	1.071)	0.481
Processed Meat	0.230	(−0.762	1.222)	0.647	0.694	(−0.593	1.981)	0.287
White Meat	0.046	(−0.598	0.690)	0.888	−0.208	(−0.917	0.499)	0.560
Fish	−0.490	(−1.106	0.126)	0.118	−0.720	(−1.383	−0.058)	0.033

Model 1: adjusted for sex and age; Model 2: adjusted for sex, age, body mass index, energy intake (kcal/day), and physical activity (METs-min/week).

**Table 3 nutrients-11-02359-t003:** Quantile regression models with Liver Iron Content as the dependent variable and dietetic factors as independent variables.

Model 1					Model 2			
	β	(95% CI)		*p*-Value	β	(95% CI)		*p*-Value
Red Meat	0.028	(−0.013	0.070)	0.176	−0.007	(−0.052	0.038)	0.753
Processed Meat	0.048	(−0.020	0.117)	0.162	0.079	(0.008	0.150)	0.028
White Meat	0.002	(−0.034	0.039)	0.882	−0.012	(−0.055	0.030)	0.570
Fish	−0.019	(−0.057	0.018)	0.315	−0.017	(−0.059	0.023)	0.395

Model 1: adjusted for sex and age; Model 2: adjusted for sex, age, body mass index, energy intake (kcal/day), and physical activity (METs-min/week).

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
