# Peer review of "Association between Different Animal Protein Sources and Liver Status in Obese Subjects with Non-Alcoholic Fatty Liver Disease: Fatty Liver in Obesity (FLiO) Study"

_nutrients, 2019, doi:10.3390/nu11102359_

Round 1
Reviewer 1 Report
This is an interesting study examining the relationship between different meat sources and NAFLD, although the study shows that there is a slight correlation between red and processed ,meat consumption and NAFLD, some of the variables analysed do not show any significant correlations- the results and the discussion sections need to be rephrased accordingly to avoid mis conclusions by the readers. Another important aspect that should be discussed here is the disease severity in the patients that were selected- was the degree of liver disease similar? How well defined was this cohort as it is a subset of a larger study.
Line 47: Obesity is a chronic disease.
Line 51: have risen
Line 56: Liver biopsies appear all of a sudden? The introduction is a mix of a lot of different things, the authors should rewrite this and discuss the different concepts clearly.
Line 67 with reference 15 is unnecessary at this place- maybe move this to where you start discussing lifestyle choices as an approach to NAFLD management.
In methods (Line 103- the authors mention processing blood- can they please elaborate on what was done?
Line 151- 159 please check the language, for example the last line should read variables associated with body composition did not “show” any statistically significant differences.
The cut offs for higher and lower meat consumption have not been explained properly, where did the authors get these thresholds from? Was it the median value for the group?
Rephrase line 182- it sounds like ferritin concentration was positively associated with red, processed and white meat, although it is a positive correlation but it is not a strong one and only red meat reaches significance, the sentence should be rephrased to avoid any mis conclusions.
The Results section needs to be rephrased as a whole, there are several statements which need to be reworded to avoid confusion.
The elastography results are interesting- even though the authors selected patients with known NAFLD they did not observe any differences in the cohorts using this technique, this needs to be highlighted and explained a bit more- were the NAFLS patients graded? What stage was this NAFLD at in the cohort selected by the authors for this study? Was there similar disease pathology?
Assessment of the serum iron is also essential as iron co-toxicity is a known factor in progression of NAFLD and liver disease in general, and liver disease also leads to changes in iron metabolism in patients, can the authors comment whether the observations made here are because of the specific meat consumption in a group or because of liver disease- i.e whether changes observed in iron concentrations are a cause or an effect?
Reviewer 2 Report
The manuscript describes the study on the association between different animal protein sources and clinical parameters related with NAFLD pathology in obese subjects. The study would be of interest to readers of this journal and the paper is generally well written. However, there were some errors or mismatches in the data description, some of which are critical in this study.
L168-170 – There was no asterisk in the liver stiffness data of participants above the median of processed meat consumption in Table 1. If the decrease was statistically significant, add an asterisk.
L178-179 – Similarly no asterisk was indicated in the liver iron data of participants with higher red meat consumption in Table 1. This point is critically important as the increased liver iron content is a key finding to consider the mechanism of the increased ALT levels in the same participants (L225-235, L256-272).
L181-183 – Processed meat and white meat were not positively correlated with serum ferritin concentration. There was no statistical significance in these comparisons (p=0.254 and p=0.869, respectively). Scientific findings should be described on the basis of statistical results.
Minor points:
L225 – Excess circulating iron is potentially toxic...
L262-263 – red and processed meat are “partly” associated with liver iron content.
